

# Investigating the eye movement characteristics of basketball players executing 3-point shot at varied intensities and their correlation with shot accuracy

Xuetong Zhao[1,*], Chunzhou Zhao[1,*], Na Liu[2] and Sunnan Li[1]

[1] College of P.E and Sports, Beijing Normal University, Beijing, China
[2] College Office, Guangdong Country Garden Polytechnic, Qingyuan, China
[*] These authors contributed equally to this work.

Corresponding author
Sunnan Li, sunnanli99@163.com

## ABSTRACT

**Background**. The 3-point shot plays a significant and pivotal role in the historical context of basketball competitions. Visual attention exerts a crucial influence on the shooting performance of basketball players. This study aims to investigate the eye movement characteristics exhibited by high-level basketball players while executing 3-point shot at varying exercise intensities, as well as explore the correlation between these eye movement characteristics and 3-point field goal percentage.

**Methods**. A total of twenty highly skilled female basketball players were recruited as participants for this study. During the experiment, the participants wore an eye tracker to record their eye movement data while executing 3-point shot at varying exercise intensities (low, moderate, and high). The collected eye movement data was analyzed using Tobii Pro Lab software. Additionally, the participants' exercise intensity was monitored by wearing Polar Team Pro sensors.

**Results**. The average number of fixations during the execution of a 3-point shot at three exercise intensities exhibited statistically significant differences in the front, bottom, top left, and bottom right. Moreover, notable disparities were observed in the average fixation duration for the front, bottom, and bottom right. The average total number of fixations and fixation duration in the moderate intensity shot were comparatively lower than those observed in the low and high intensity shots, while the average number of fixations and percentage of fixation duration on the front were relatively higher compared to those in the low and high intensity shots. Under varying intensities, there were no significant differences observed in the average number of fixations and the 3-point field goal percentage each AOI; however, a significantly positive correlation was found between the front average fixation duration and the 3-point field goal percentage.

**Conclusion**. During the execution of a moderate intensity 3-point shot, the player's fixation is focused and stable, their information search strategy is efficient, and their information processing is precise. Variations in exercise intensity result in changes to both the information search strategy and degree of processing. Fixating on the front has a positive impact on 3-point field goal percentage.

## INTRODUCTION

Eye tracking technology is a highly effective and real-time method for accurately capturing temporal and spatial features, enabling the exploration of cognitive processing. Visual fixation characteristics are assessed through measurements of fixation, saccadic amplitude, and regression in response to visual stimuli (*Conklin, Pellicer-Sánchez & Carrol, 2020*). In recent years, there has been an increasing utilization of eye tracking technology among researchers to investigate the visual attention characteristics exhibited by basketball players (*Zhao, Liu & Li, 2024*; *Zhao, Li & Zhao, 2023*; *Jin et al., 2020*).

With the advancement of basketball competition level, there has been a gradual improvement in shooting accuracy among basketball players, accompanied by an increase in shooting distance. The 3-point shot holds a significant historical and pivotal role in basketball competitions (*Gould et al., 2014*). Nowadays, an increasing number of teams are prioritizing the training and enhancement of their 3-point shooting skills due to the fact that while an average of 100 2-point shots yields only 79 points for a team, taking 100 3-point shots results in an average of 105 points (*Shea, 2020*). The Golden State Warriors' extraordinary 73-win and 9-loss season in 2015–2016 can primarily be attributed to their astute utilization of the 3-point shot as a strategic advantage against opponents (*Young, 2019*). Given the pivotal role of the 3-point shot in securing triumphs, basketball coaches and trainers employ diverse methodologies to optimize players' performance in this aspect, including enhancing training velocity (*Lu et al., 2020*). Visual fixation serves as the primary modality through which participants acquire decision-making information, and the number fixations reflects their individualized information search strategy (*Conklin, Pellicer-Sánchez & Carrol, 2020*). The ability to control attention range and direction serves as a crucial indicator of basketball players' psychological abilities, directly influencing their players' performance (*Sun, 2004*). *Jiang (2000)* believes that visual attention is very important in basketball shooting and is a prerequisite for the formation of good muscle proprioception and improved shooting rates. Attention is widely recognized as a crucial factor influencing motor learning and performance outcomes (*Wulf & Prinz, 2001*; *Xu, 2012*). Specifically, attention is considered as the cognitive process responsible for information detection and processing (*Moeinirad et al., 2020*), with external attentional focus being found to significantly impact optimal performance levels (*Lewthwaite & Wulf, 2017*; *Wulf & Lewthwaite, 2016*). Basketball players with a high shooting percentage move their fixation appropriately to discover key visual information that may predict a shot (*Mizuguchi, Honda & Kanosue, 2013*). Visual control training has been shown to be effective in enhancing motor learning and performance (*Oudejans, 2012*; *Sun, 2004*), while the utilization of software and navigator tools has also been explored (*Guo, Li & Xin, 2012*). Experienced players adeptly shift their gaze in response to the shooter's movements, thereby discerning crucial visual cues that may forecast shot success (*Mizuguchi, Honda & Kanosue, 2013*). In the context of basketball players, attention serves as a prerequisite

for developing accurate proprioception and enhancing shooting accuracy (*Sun & Fu, 2010*). Previous studies have demonstrated the impact of physical exercise on information processing and cognition (*Tomporowski & Ellis, 1986*). Basketball is a physically demanding sport characterized by its competitive nature. Special endurance is crucial for sustaining optimal skill performance and physical fitness over extended periods, encompassing both maximum and sub-maximum intensity loads. The core value of load intensity lies in its close association with competition intensity (*Tian, 2017*). The improvement of players' competitive performance ability, based solely on the competition load characteristics of an event, holds significant practical implications for training under a specific intensity. According to *Zhang & Zhao (2005)*, high-intensity exercise constitutes 41% of the total activity in a high-level basketball game, while moderate-intensity exercise accounts for 44%, and low-intensity exercise makes up the remaining 15%. While maintaining a high percentage of successful 3-point shots during regular training or low-intensity activities is feasible for basketball players, it becomes challenging to sustain such accuracy as exercise intensity increases, particularly during the second half of games. The findings suggest that physical exertion may reduce oculomotor efficiency during aiming at a distant target. Moreover, stationary and dynamic shots require different gaze behavior strategies (*Zwierko et al., 2018*).

Currently, researchers and practitioners are increasingly recognizing the detrimental effects of mental fatigue resulting from high-intensity training and competition on athletes' cognitive and sports performance (*Mansec et al., 2018*; *Smith et al., 2018*). Previous studies have demonstrated that mental fatigue impairs executive control, leading to a detrimental impact on task performance (*Kato, Eendo & Kizuka, 2009*). *Boksem, Meijman & Lorist (2005)* found that mental fatigue reduces subjects' attentional control and inhibitory abilities towards irrelevant stimuli. Additionally, enhanced arousal levels during moderate intensity exercise led to accelerated speed of cognitive processing, thereby enhancing exercise performance (*McMorris & Hale, 2012*). Studies have demonstrated that the activation of the locus coeruleus, mediated by feedback from stretch reflexes, baroreceptors, and β-adrenoceptors on the vagus nerve beyond the post-catecholamines threshold, can account for the effects of moderate exercise. The facilitation of various tasks is achieved through the stimulation of the reticular system by norepinephrine (NE). However, central executive tasks are further enhanced by the activation of α2A-adrenoceptors and D1-dopaminergic receptors in the prefrontal cortex. This activation increases the signal to 'noise' ratio, thereby improving motor performance (*McMorris, 2016*). From the perspective of limited cognitive resources, the decline in motor performance primarily stems from excessive occupation of cognitive resources by task-irrelevant information, resulting in insufficient allocation of cognitive resources to the target task and thereby impeding focused attention on task-related information processing. Specifically, the decline in motor performance is frequently accompanied by a concomitant decrease in attentional control, an augmented allocation of attention towards irrelevant or threatening information, a reduced duration of attention on the target stimulus, and an increased difficulty in disengaging attention from distracting information (*Ai & Zhang, 2018*).

The utilization of eye trackers plays a pivotal role in investigating the collective fixation strategy among basketball players (*Gong et al., 2016*). However, current studies of this nature predominantly involve participants in a state of tranquility and employ visual stimuli, such as pictures or video scenes depicting sports practice, as experimental materials to explore their fixation patterns and information processing efficiency (*Wang, Li & Yan, 2007*; *Li, Xu & Zhang, 2006*). Previous studies have investigated the gaze characteristics of basketball players during shooting, but they did not consider exercise intensity as an intervening factor (*Zhao, Li & Zhao, 2023*; *Gou, Li & Wang, 2022*). Building upon the aforementioned findings, this study incorporates exercise intensity as an independent variable and investigates the fixation characteristics and shooting accuracy of basketball players during 3-point shot attempts under varying exercise intensities, while also exploring their interrelationships. (1) The fixation characteristics of players when executing 3-point shot vary significantly based on different exercise intensities; (2) fixation characteristics demonstrate a significant correlation with 3-point field goal percentage under different intensities. This study furnishes a scientific foundation for basketball shooting pedagogy and training.

## MATERIALS AND METHODS

### Participants

By consulting previous studies of scholars, the ideal statistical test power and effect size should be higher than 0.8, and the two-tailed α significance level should be 0.05 (*Cohen, 1988*; *Faul et al., 2007*; *Faul et al., 2009*). Using this as the standard, G*Power 3.1.9.7 (Germany) software is used to estimate that the total sample size required in the single-factor 3-level within-group design is 20, which can meet the experimental requirements (*Faul et al., 2007*; *Faul et al., 2009*).To account for possible attrition effects, we ultimately determined that our final sample should consist of 22 subjects (*Jin et al., 2020*). These recruited individuals subsequently completed the 3-point shot task across varying exercise intensities: low intensity, moderate intensity, and high intensity. All participants were selected from the women's basketball team of Beijing Normal University (mean age = 23 ± 2.60). All participants had achieved victory in the China University Women's Basketball League championship, with an additional five participants having triumphed in the World University Women's Basketball Championship held in 2023. Furthermore, it is noteworthy that all participants exhibited right-handed shooting proficiency. This study adhered to the ethical standards for human subject research and received approval from the Institutional Review Board (IRB-20221126) at the School of Physical Education and Sports, Beijing Normal University. Prior to participating in the experiment, each participant provided written informed consent.

### Apparatus

The experiment utilized the Tobii Glasses 3 portable eye tracker, manufactured in Sweden, with a sampling rate of 100 Hz. This device does not impede the wearer's field of vision and allows for unrestricted head and body movements while ensuring high-quality data acquisition, thus capturing natural and authentic behavior. During the experiment,

participants wore wearable eye-tracking glasses and a storage device. The wearable eye-tracking glasses were primarily utilized for capturing participants' eye movement data, while the storage device served as a means of storing such data.

The eye movement data of the participants was analyzed using Tobii Pro Lab (version 1.21.21571) software. Before the analysis of the eye movement data, the memory card was initially detached from the storage device, followed by launching the Tobii Pro Lab software on the computer. Subsequently, in accordance with experimental requirements, the eye movement data recorded on the memory card were imported into the software for analysis. The Analysis menu was primarily utilized to complete fixation duration, number of fixations, and other data, while the visualization menu facilitated visual analysis. After completing the analysis, the raw data was exported to Excel format in accordance with the requirements of experimental analysis (*Niehorster, Hessels & Benjamins, 2020*). The Polar Team Pro sensor, manufactured in Finland, was utilized for monitoring participants' heart rate during various exercise intensities.

## Exercise intensity division

According to the standard of human body's response to exercise load and heart rate, it is commonly categorized into four levels of intensity: high intensity, defined as a heart rate exceeding 157 beats per minute; moderate intensity, ranging from 156 to 139 beats per minute; low intensity, ranging from 138 to 120 beats per minute; and general activities, characterized by a heart rate below 120 beats per minute (*Zhao & Zhao, 2005*). In this experiment, based on the exercise intensity grading standard proposed by *Chen, Yin & Yan (2011)* and considering the actual intensity observed in basketball games, we assessed participants' shooting performance at three different levels of intensity: low (120–138 times/min), medium (139–156 times/min), and high (157 times/min).

## Design of experiment

The present study employed a within-group design. The independent variable was exercise intensity, which was classified into three levels (low, moderate, and high). The dependent variables were eye movement indices, including the number of fixations, fixation duration, and 3-point field goal percentage. The experiment was conducted at the basketball court of Beijing Normal University Gymnasium. During different exercise intensities (low, moderate, and high), participants were equipped with an eye tracker while executing 3-point shots. Each participant completed three rounds of 3-point shots at varying intensities, and the corresponding data were recorded.

## Experimental procedure

The experiment was concluded on September 9, 2023, and it was conducted under the experimenter. Prior to commencing the experiment, all participants were provided with a comprehensive explanation of the experimental knowledge, requirements, and precautions. Simultaneously, all participants were equipped with heart rate sensors, and subsequently, based on the draw order, one participant entered the basketball court to engage in activities that increased exercise intensity through running back, dribbling, and shooting. The Polar Team Pro sensors were employed for monitoring participants' heart rates while the

remaining participants awaited in the rest room. Upon reaching the designated intensity of heart rate, the experimenter guided the participant to the testing area and facilitated the placement of an eye tracker (refer to Fig. 1A). Following calibration, the experimental trial commenced. Each participant executed three shots at each level of intensity (the video footage from the initial trial was utilized for data analysis, while subsequent trials served as backups), with Fig. 1 illustrating shot placements. The order was counterbalanced to mitigate the potential influence of test order effects, with each exercise intensity test being re-administered. The sequence of exercise intensities consisted of low, medium, and high levels, with an average duration of approximately 20 min per participant. The entire experimental procedure lasted for approximately 400 min.

## Area of interest (AOI)

The AOI refers to the specific area that participants focus on during information search. To establish AOIs for this study, consultations were conducted with basketball experts and instrument manufacturer professionals, taking into account the participants' fixation areas during the 3-point shot (refer to Fig. 1B).

## Three-point field goal percentage test

The testing period is scheduled to take place from September 10th to 14th, 2023, with the designated venue being the basketball court located within Beijing Normal University Gymnasium. All participants are mandated to partake in the examination. One hundred shots were tested at each intensity, with the success and failure of each shot recorded to calculate the final percentage. (The average hit rate for low intensity was 52.15%, moderate intensity was 63.25%, while high intensity was 43.3%). The test was conducted ten times in both the morning and afternoon for five consecutive days using the same method, sequential arrangement, and intensity as in the eye movement experimental study.

## Eye tracking data

Fixation was operationally defined as a state in which the eye remained stationary within a tolerance of movements up to 3° for at least 100 ms (*Panchuk & Vickers, 2006*), while fixation duration referred to the total time that gaze remained fixed within an area of interest during a given interval (Tobii Pro Lab User Manual). Fixation count represented the number of fixations occurring within this particular area of interest during an interval (Tobii Pro Lab User Manual).

## Data analysis

The eye movement data of the participants during the 3-point shot were analyzed using Tobii Pro Lab (version 1.21.21571). The collected data were exported to SPSS 26.0 in EXCEL format for statistical analysis and ANOVA. In case of a significant effect, post-hoc LSD tests were conducted for multiple comparisons. The significance level was set at $p < 0.05$. The Spearman's correlation coefficient was computed to assess the relationship between the number of fixations and the 3-point field goal percentage. The magnitude of each correlation coefficient was evaluated using the following criteria: <0.1 = trivial; 0.1–0.3 = small; 0.3–0.5 = moderate; and 0.5–0.7 = large (*Hopkins et al., 2009*).

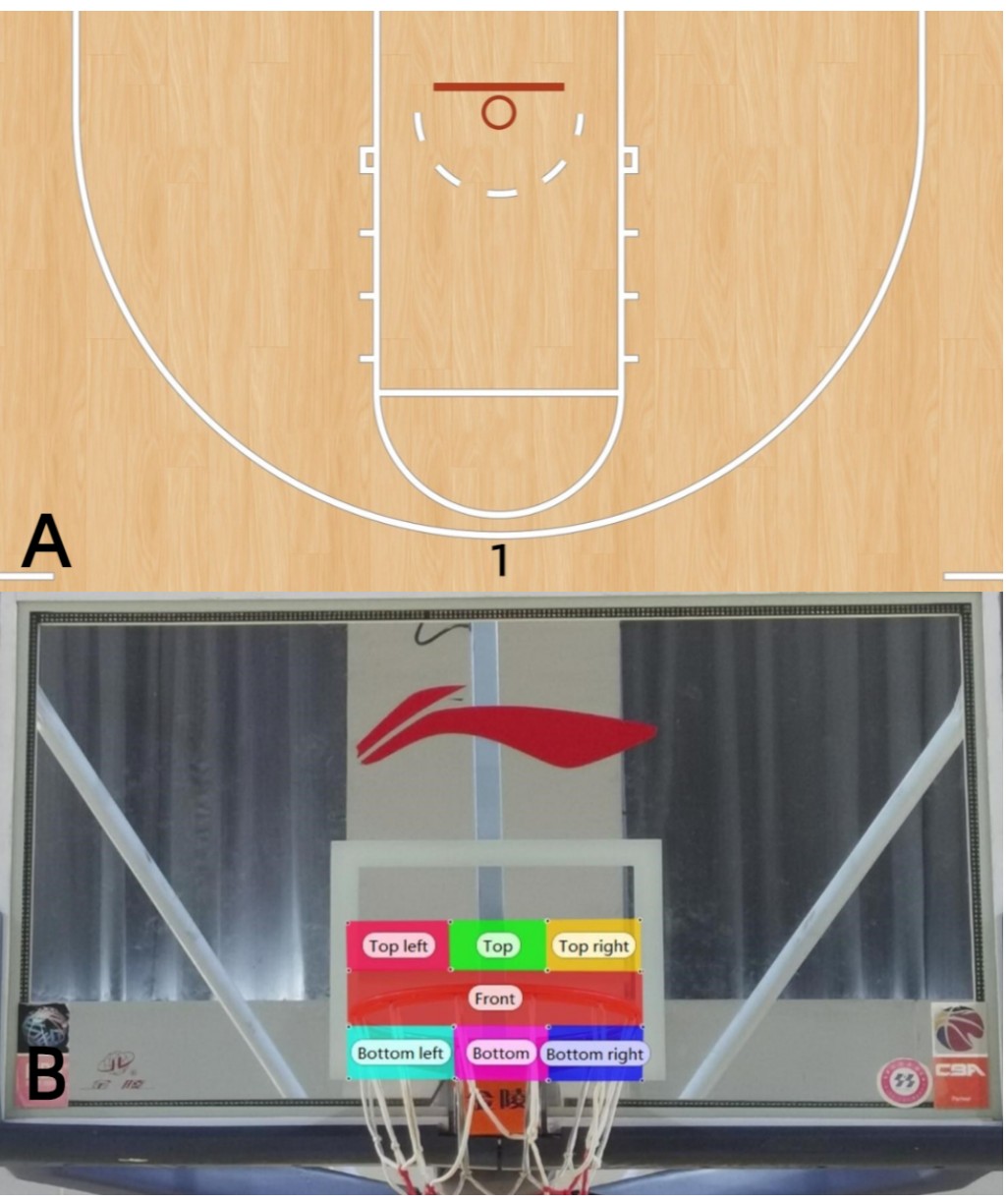

**Figure 1    Shooting position and AOI.**

## RESULTS

### AOI number of fixations

In order to investigate the disparity in the mean number of fixations among participants on each AOI while performing 3-point shots under varying exercise intensities, an analysis of variance (ANOVA) was conducted with exercise intensity as the independent variable and the number of fixations on each AOI as the dependent variable (refer to Table 1). Following a K-S test, it was determined that the number of fixations for all participants during 3-point shot execution under different exercise intensities exhibited a normal distribution.

**Table 1  Comparison of number of fixations in different AOI under different intensities.**

| AOI | Low intensity | | Moderate intensity | | High Intensity | | $F$ | $P$ value |
|---|---|---|---|---|---|---|---|---|
| | M | SD | M | SD | M | SD | | |
| Front | 1.90 | 0.55 | 1.60 | 0.50 | 2.25 | 0.72 | 5.929 | 0.005[*] |
| Top | 1.95 | 0.60 | 1.75 | 0.55 | 2.00 | 0.65 | 0.964 | 0.388 |
| Bottom | 1.35 | 0.49 | 0.75 | 0.94 | 1.85 | 0.67 | 16.585 | <0.001[**] |
| Top left | 1.55 | 0.60 | 1.35 | 0.93 | 1.75 | 0.72 | 1.371 | 0.262 |
| Bottom left | 0.60 | 0.60 | 0.50 | 0.61 | 1.00 | 0.79 | 3.093 | 0.053 |
| Top right | 1.40 | 0.60 | 1.30 | 0.47 | 1.80 | 0.70 | 3.950 | 0.025[*] |
| Bottom right | 0.55 | 0.51 | 0.25 | 0.44 | 0.90 | 0.55 | 8.321 | 0.001[*] |

Notes.
[*]The mean difference is significant at the 0.05 level.
[**]The mean difference is significant at the 0.01 level.

As depicted in Table 1, regarding the main effect of exercise intensity, a significant difference was observed in the front ($F_{(2, 57)} = 5.929$, $P = 0.005$, $\eta2 = 0.172$). Subsequent post-hoc LSD analysis revealed a statistically significant disparity in the average number of fixations between moderate and high-intensity 3-point shot ($P < 0.001$). The bottom exhibited a significant main effect of exercise intensity ($F_{(2, 57)} = 16.585$, $P < 0.001$, $\eta2 = 0.368$). Subsequent post-hoc LSD analysis revealed a statistically significant difference in the average number of fixations between the low and moderate intensity ($P = 0.003$). Furthermore, there was a significant disparity in the average number of fixations between the low and high-intensity ($P = 0.011$). The top right exhibited a significant main effect of exercise intensity ($F_{(3, 76)} = 3.950$, $P = 0.025$, $\eta2 = 0.122$). Subsequent post-hoc LSD analysis revealed statistically significant differences in average number fixations among high intensity, low intensity ($P = 0.038$), and moderate intensity ($P = 0.010$). The bottom right exhibited a significant main effect of exercise intensity $F_{(2, 57)} = 8.321$, $P = 0.001$, $\eta2 = 0.226$. Subsequent *post-hoc* LSD tests indicated statistically significant differences ($P = 0.032$) in the average number of fixations between high intensity and moderate intensity as well as moderate intensity.

### AOI fixation duration

In order to investigate the disparity in average fixation duration across each AOI when participants attempt a 3-point shot under varying exercise intensities, an analysis of variance (ANOVA) was conducted with exercise intensity as the independent variable and fixation duration within each AOI as the dependent variable (refer to Table 2). Following a Kolmogorov–Smirnov test, it was determined that the fixation duration data for all participants while shooting 3-point shots under the three exercise intensities exhibited normal distribution. As depicted in Table 2, regarding the main effect of exercise intensity, a significant difference was observed in the front ($F_{(2, 57)} = 31.076$, $P < 0.001$, $\eta2 = 0.521$). Subsequent post-hoc LSD analysis revealed statistically significant differences in average fixation duration between conditions of low and moderate intensity as well as high intensity ($P < 0.001$). The bottom exhibited a significant main effect of exercise intensity ($F_{(2, 57)} = 70.283$, $P < 0.001$, $\eta2 = 0.711$). Subsequent post-hoc LSD analysis revealed

**Table 2 Comparison of fixation duration in different AOI under different intensities (ms).**

| AOI | Low intensity | | Moderate intensity | | High Intensity | | F | P value |
|---|---|---|---|---|---|---|---|---|
| | M | SD | M | SD | M | SD | | |
| Front | 611 | 100 | 789 | 59 | 583 | 99 | 31.016 | <0.001** |
| Top | 393 | 93 | 426 | 71 | 405 | 93 | 0.707 | 0.497 |
| Bottom | 464 | 62 | 164 | 132 | 485 | 80 | 70.283 | <0.001** |
| Top left | 278 | 99 | 301 | 182 | 326 | 99 | 0.650 | 0.526 |
| Bottom left | 113 | 116 | 90 | 108 | 164 | 125 | 2.085 | 0.134 |
| Top right | 307 | 95 | 359 | 72 | 372 | 112 | 2.626 | 0.081 |
| Bottom right | 115 | 98 | 38 | 70 | 143 | 77 | 8.598 | 0.001* |

**Notes.**
*The mean difference is significant at the 0.05 level.
**The mean difference is significant at the 0.01 level.

statistically significant differences in the average fixation duration among low intensity, moderate intensity, and high intensity conditions ($P < 0.001$). The main effect of exercise intensity was found to be significant in the bottom right, ($F_{(2, 57)} = 8.598$, $p = 0.001$, $\eta2 = 0.232$). Subsequent LSD analysis indicated a significant difference in average fixation duration between high intensity and moderate intensity ($P < 0.001$).

## Number of fixations and fixation duration distribution

As illustrated in Figs. 2 and 3, participants demonstrated the lowest average total number of fixations (7.5 times) and total fixation duration (2,167 ms) during moderate-intensity 3-point shots, while the highest average total number of fixations (11.55 times) and fixation duration (2,478 ms) were observed during high-intensity 3-point shots. Moderate intensity fixation was primarily concentrated on the front (21%), top (23%), top left (18%), and top right (17%). Low intensity fixation was mainly distributed in the front (20%), top (21%), bottom (14%), top left (17%) and top right (15%). High intensity fixation was predominantly distributed in the front (19%), top (17%), bottom (16%), top left (15%) and top right (15%). The statistical findings clearly demonstrate that participants displayed a narrower distribution of fixations at moderate intensity in comparison to low and high intensities, with the highest range of fixations observed at high intensity. Notably, during moderate intensity, the average fixation duration in front was significantly longer and accounted for the largest proportion of the average total fixation duration (36%). Simultaneously, the distribution of fixation encompassed the area surrounding and above the basket for shots of moderate intensity, while also occupying a significant proportion below the basket for both low and high-intensity shots. This observation indicates that variations in exercise intensity exert a substantial influence on participants' fixation distribution during shooting.

## Spearman correlation between the number of fixations and 3-point field goal percentage

Table 3 shows that at low intensity, a small negative correlation was observed between the front position ($r = -0.293$, $p = 0.211$) and the 3-point field goal percentage. Additionally,

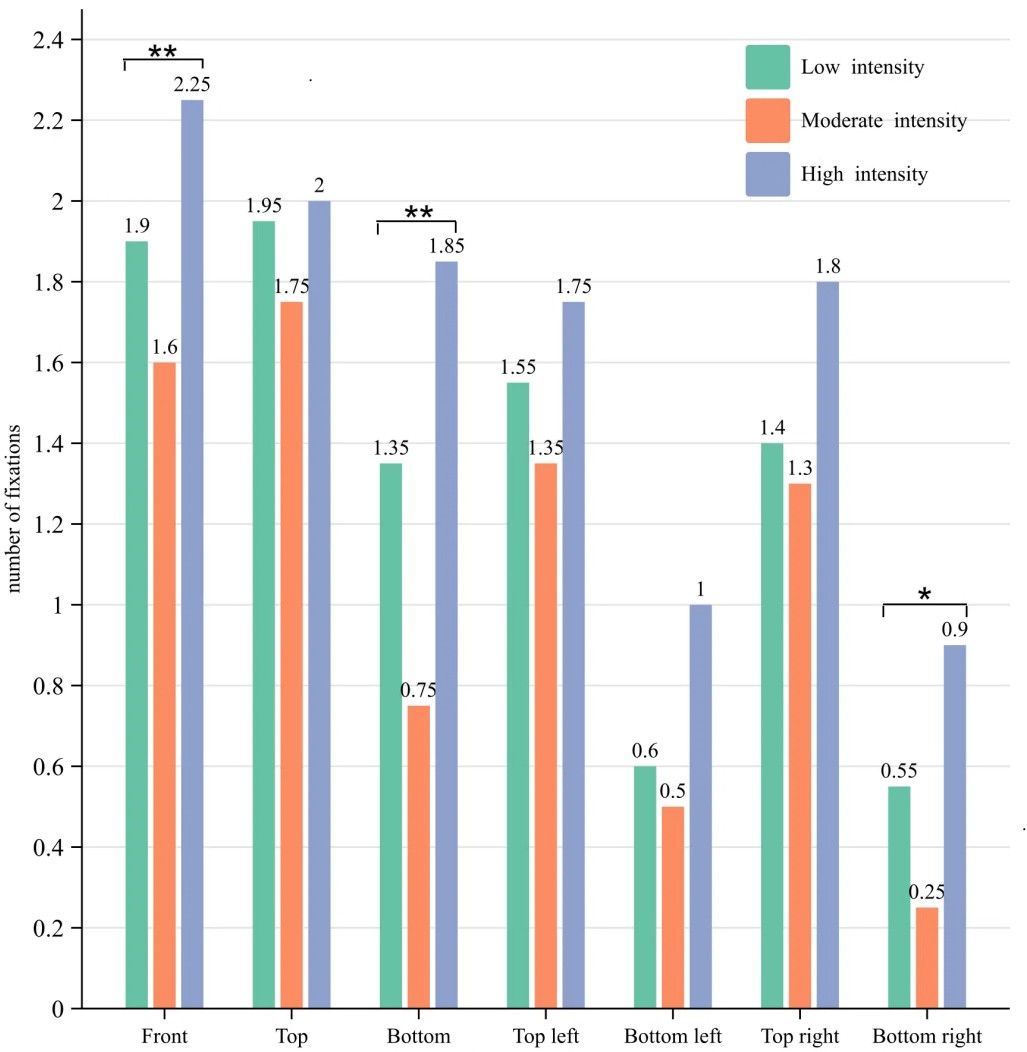

**Figure 2** **Distribution of the number of fixations.** Notes: The average total number of fixations were 9.3 for low intensity, 7.5 for moderate intensity, and 11.55 for high intensity.

there were small positive correlations between the top ($r = 0.129$, $p = 0.589$), top left ($r = 0.184$, $p = 0.438$), bottom left ($r = 0.157$, $p = 0.510$), and the 3-point field goal percentage; a trivial positive correlation was observed between the bottom ($r = 0.060$, $p = 0.801$) and the 3-point field goal percentage; a trivial negative correlation was observed between the top right ($r = -0.081$, $p = 0.753$) and 3-point field goal percentage, while a small negative correlation was observed between the bottom right ($r = -0.167$, $p = 0.481$) and the 3-point field goal percentage. At moderate intensity, there were trivial positive correlation observed between the front ($r = 0.090$, $p = 0.706$), top ($r = 0.062$, $p = 0.796$), bottom ($r = 0.077$, $p = 0.748$), top right ($r = 0.016$, $p = 0.947$), and bottom right ($r = 0.025$, $p = 0.951$) with the 3-point field goal percentage; a moderate positive correlation was observed between the bottom left ($r = 0.385$, $P = 0.094$) with the 3-point

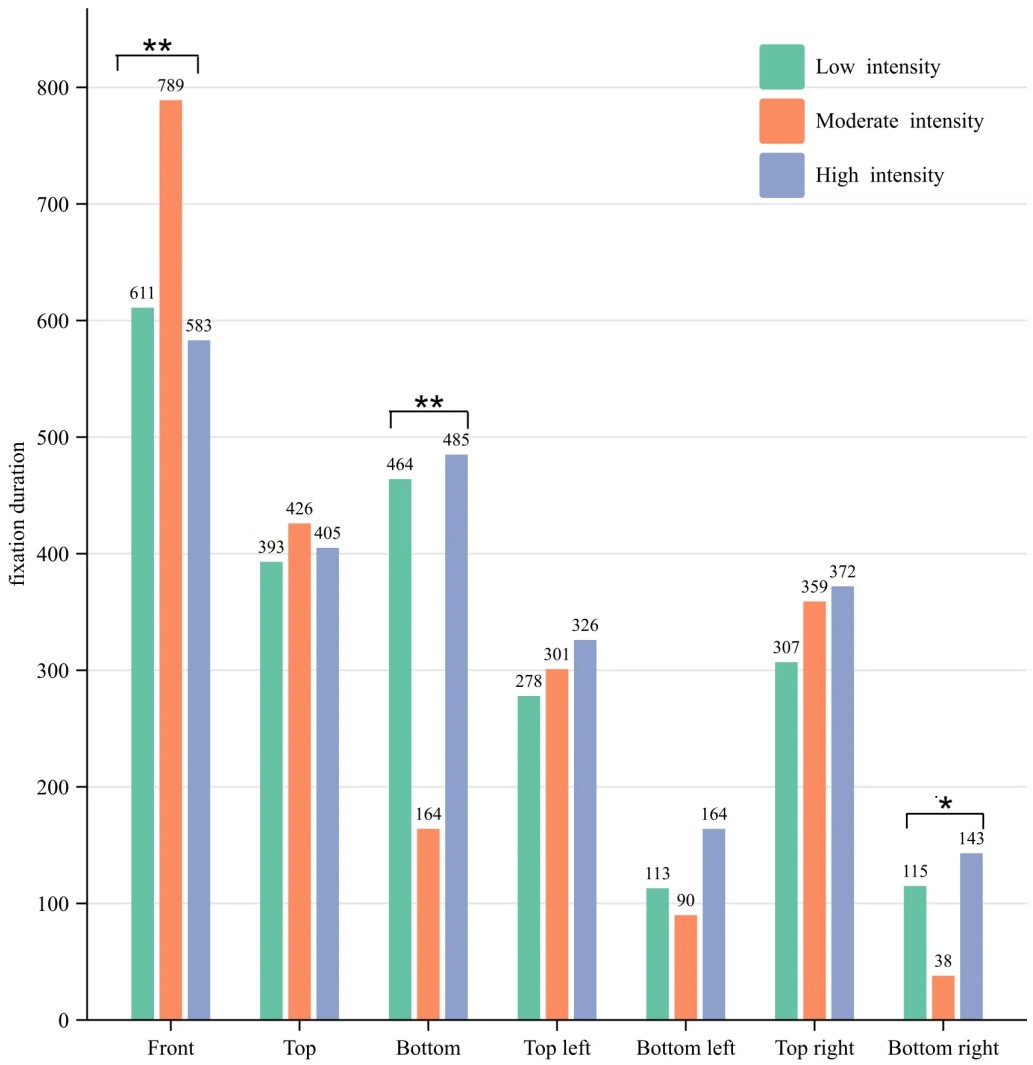

**Figure 3** **Distribution of the fixation duration.** Notes: the average total fixation duration was 2281 for low intensity, 2167 for moderate intensity, and 2478 for high intensity.

field goal percentage; while a trivial negative correlation was observed between the top left ($r = -0.012$, $p = 0.960$) and the 3-point field goal percentage. At high intensity, there were small negative correlations was observed between the front ($r = 0.147$, $p = 0.536$), top ($r = -0.108$, $p = 0.649$), and bottom right ($r = -0.163$, $p = 0.493$) with the 3-point field goal percentage; a moderate negative correlation ($r = -0.379$, $p = 0.099$) was observed between the bottom and the 3-point field goal percentage; a trivial positive correlation was observed between the top left ($r = 0.069$, $p = 0.774$) and the 3-point field goal percentage; a small positive correlation was observed between the bottom left ($r = 0.177$, $p = 0.456$) and the 3-point field goal percentage; a trivial positive correlation was observed between the top right ($r = 0.097$, $p = 0.684$) and the 3-point field goal percentage;

**Table 3** Correlation between number of fixations and 3-point field goal percentage ($N = 20$).

| AOI | Low intensity | | Moderate intensity | | High intensity | |
|---|---|---|---|---|---|---|
| | $r$ | $p$ | $r$ | $p$ | $r$ | $P$ |
| Front | −0.293 | 0.211 | 0.090 | 0.706 | −0.147 | 0.536 |
| Top | 0.129 | 0.589 | 0.062 | 0.796 | −0.108 | 0.649 |
| Bottom | 0.060 | 0.801 | 0.077 | 0.748 | −0.379 | 0.099 |
| Top left | 0.184 | 0.438 | −0.012 | 0.960 | 0.069 | 0.774 |
| Bottom left | 0.157 | 0.510 | 0.385 | 0.094 | 0.177 | 0.456 |
| Top right | −0.081 | 0.735 | 0.016 | 0.947 | −0.097 | 0.684 |
| Bottom right | −0.167 | 0.481 | 0.025 | 0.915 | −0.163 | 0.493 |

### Spearman correlation between the fixation duration and 3-point field goal percentage

Table 4 shows that at low intensity, a significant positive correlation was observed between the fixation duration on front ($r = 0.794$, $p < 0.001$) and the 3-point field goal percentage (see Fig. 4); a trivial positive correlation between the top ($r = 0.063$, $p = 0.792$) and the 3-point field goal percentage; a trivial negative correlation between the top left ($r = -0.051$, $p = 0.832$) and the 3-point field goal percentage; a small positive correlation between the bottom left ($r = 0.113$, $p = 0.637$) and the 3-point field goal percentage; there were small negative correlations between the bottom ($r = -0.182$, $p = 0.444$), top right ($r = -0.126$, $p = 0.596$), and bottom right ($r = -0.291$, $p = 0.231$) with the 3-point field goal percentage. At moderate intensity, a significant positive correlation was observed between the front ($r = 0.649$, $P = 0.002$) and the 3-point field goal percentage (see Fig. 4); there were trivial negative correlations between the top ($r = -0.087$, $p = 0.717$), top left ($r = -0.002$, $p = 0.995$), top right ($r = -0.064$, $p = 0.789$), and bottom right ($r = -0.054$, $p = 0.822$) with the 3-point field goal percentage; a small negative correlation between the bottom ($r = -0.129$, $p = 0.589$) with the 3-point field goal percentage; a moderate positive correlation between the bottom left ($r = 0.412$, $p = 0.071$) with the 3-point field goal percentage. At high intensity, a significant positive correlation ($r = 0.625$, $P = 0.003$) was observed between the front and the 3-point field goal percentage (see Fig. 4); a trivial positive correlation between the top ($r = 0.051$, $p = 0.832$) with the 3-point field goal percentage; a trivial negative correlation between the top left ($r = -0.098$, $p = 0.682$) with the 3-point field goal percentage; there were small negative correlations between the top right ($r = -0.257$, $p = 0.274$), and bottom right ($r = -0.236$, $p = 0.316$) with the 3-point field goal percentage; a moderate positive correlation between the bottom left ($r = 0.300$, $p = 0.199$) with the 3-point field goal percentage; a moderate negative correlation between the bottom ($r = -0.344$, $p = 0.138$) with the 3-point field goal percentage.

**Table 4** Correlation between fixation duration and the 3-point field goal percentage ($N = 20$).

| AOI | Low intensity | | Moderate intensity | | High intensity | |
|---|---|---|---|---|---|---|
| | r | p | r | p | r | P |
| Front | 0.794[**] | <0.001 | 0.649[**] | 0.002 | 0.625[*] | 0.003 |
| Top | 0.063 | 0.792 | −0.087 | 0.717 | 0.051 | 0.832 |
| Bottom | −0.182 | 0.444 | −0.129 | 0.589 | −0.344 | 0.138 |
| Top left | −0.051 | 0.832 | −0.002 | 0.995 | −0.098 | 0.682 |
| Bottom left | 0.113 | 0.637 | 0.412 | 0.071 | 0.300 | 0.199 |
| Top right | −0.126 | 0.596 | −0.064 | 0.789 | −0.257 | 0.274 |
| Bottom right | −0.291 | 0.213 | 0.054 | 0.822 | −0.236 | 0.316 |

Notes.

[*]Correlation is significant at the 0.05 level (2-tailed).

[**]Correlation is significant at the 0.01 level (2-tailed).

# DISCUSSION

## Spatial dimension eye movement metrics

When shooting 3-point shot at moderate intensity, participants mainly focused on the front of and the top the basket, However, with the variations in exercise intensity, the percentage of the average number of fixations in the front and top decreased, while the proportion of fixations in other AOI increased, indicating that the participants' number of fixations were more concentrated when shooting 3-point shot at moderate intensity than those in low intensity and high intensity. Moreover, in terms of visual fixation, participants exhibited heightened concentration during 3-point shot of moderate intensity as compared to both low and high intensity scenarios. The rationale behind this phenomenon lies in the fact that exercise intensity induces both physical and psychological fatigue (*Zhao, Li & Zhao, 2023*). When individuals are fatigued, their ability to initiate attention control may be significantly impaired, leading them to rely more on reactive control (*Li, Zhang & Qu, 2019*). Low exercise intensity, based on deep special theoretical knowledge, automatic movement skills, rich experience accumulation and stable fixation area and fixation characteristics formed in the process of long-term training, players will search for information in accordance with the habit of fixation pattern for each shot, showing attention, fixation target is clear, and has clear direction and concentration (*Zhao, Li & Zhao, 2023*; *Zhao & Li, 2023a*; *Zhao & Li, 2023b*). At moderate intensity, bodily functions are optimally activated, resulting in improved performance during shooting compared to low-intensity conditions. The visual system rapidly localizes the specific area of interest within the target range, extracts pertinent information, and subsequently generates a more rational response following processing, thereby exemplifying an efficient strategy for targeted information fixation. The target fixation strategy can be elucidated by the visual index theory, which posits that the human visual system is capable of discerning stimuli in the visual field as distinct entities through fundamental automatic operations. Moreover, this index maintains its consistency with the object despite environmental changes (*Ripoll, Bard & Paillard, 1986*).

However, as exercise intensity increases and fatigue accumulates, the efficiency of this automation diminishes. The participants' effective target fixation strategy and information

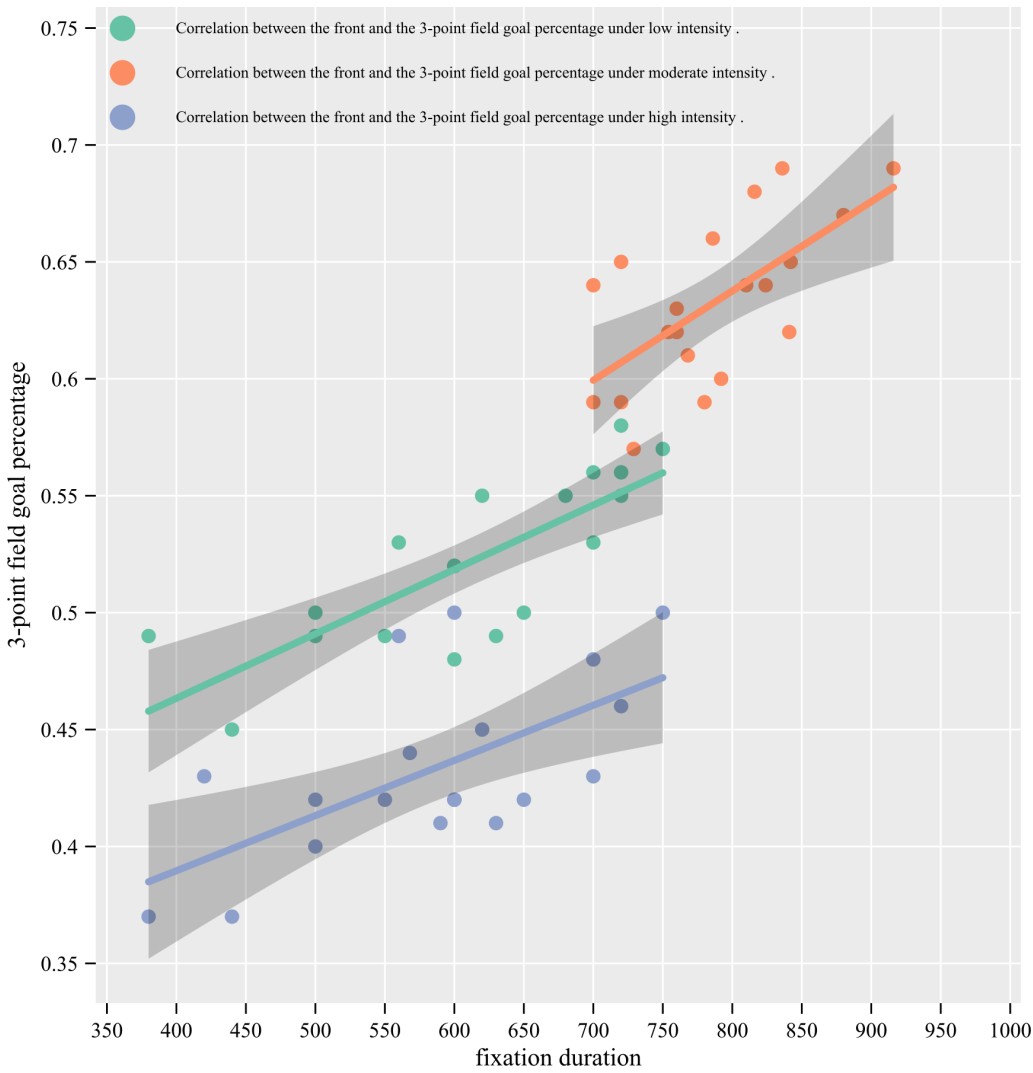

**Figure 4 Correlation between fixation duration and the 3-point field goal percentage.**

search strategy when shooting a 3-point shot at low and moderate exercise intensities can be attributed to their superior ability to encode and process specific information, as well as their enhanced capacity to predict the ball's final landing point. These advantages stem from the extensive cognitive information base (*Willams et al., 1994*; *Zhao & Li, 2023a*; *Zhao & Li, 2023b*). The player's fixation pattern exhibits greater stability when executing 3-point shot at moderate intensity compared to low intensity; however, as intensity increases to high levels, significant changes occur in the player's fixation pattern. The primary factor lies in the elevation of exercise intensity, resulting in increased energy expenditure, accumulation of fatigue, elevated heart rate, diminished mental stability, and other contributing factors that collectively impede players' active control ability and divert their attention. The direction and concentration of fixation were disrupted, thereby disrupting the original stable fixation pattern. Simultaneously, the automatic calculation ability of the visual

system was diminished, leading to a reduction in players' information search strategy while executing the 3-point shot. The findings of this study are in line with the outcomes reported in previous research (*Zhang, 2008*; *Zhao, Li & Zhao, 2023*).

## Temporal eye movement metrics

Longer fixation durations are associated with increased stability of fixation and enhanced precision in information processing (*Zhao & Li, 2023a*; *Zhao & Li, 2023b*). The level of stability is affected by various individual factors including physical fatigue, reduced interest in activities, or compromised willpower (*Xu, 2012*). The fixation duration not only reflects the temporal extent of players' engagement with each fixation position but also signifies the depth of scene information processing (*Jin, 2020*). Prior to shooting, players tend to exhibit heightened focus, thereby facilitating more refined information processing (*Zhang et al., 2004*; *Zhu, Gao & Huang, 2014*). The participants demonstrated the longest average fixation duration during moderate intensity, followed by low intensity, and the shortest average fixation duration during high intensity. Notably, the lowest average fixation duration was observed at the bottom and bottom right regions of moderate intensity. The average total fixation duration varied with exercise intensity, and the proportion of the average fixation duration allocated to each AOI within the overall average fixation duration also exhibited variation. The prolonged fixation duration was not exclusively dedicated to refining specific target regions but encompassed processing other regions as well. The findings demonstrate that exercise intensity diminishes the efficacy of information processing and exerts a significant impact on the extent of information processing. Additionally, cognitive processing factors influence the fixation duration, which is regulated by the cognitive system in controlling eye movements (*Gou, Li & Wang, 2022*; *Jin, Ge & Fan, 2023*).

At moderate intensity 3-point shot, the average fixation duration of the front and top areas accounted for the highest proportion, while also exhibiting the highest 3-point field goal percentage. As exercise intensity varied, there was a decrease in average fixation duration within these two areas, accompanied by an increase in fixation duration within other areas. Consequently, the 3-point field goal percentage gradually declined. In general, a decrease in cognitive processing load within the fixative range is associated with a reduction in fixation point duration; conversely, an increase in cognitive processing load leads to an elongation of the fixation point duration (*Hong, Liu & Li, 2007*). The vary in exercise intensity leads to a significant alteration in fixation stability, accompanied by a notable shift in the degree of information processing. This phenomenon may be attributed to the heightened energy consumption among players, resulting in the accumulation of fatigue and subsequently imposing a substantial psychological burden on individuals. In order to ensure a high 3-point field goal percentage, players strive to maintain focus on the target position. However, due to the impact of exercise intensity, participants may experience difficulty in accurately processing information related to the target area. This observation further emphasizes the significant influence of players' fixation target selection and information processing abilities on their overall 3-point field goal percentage (*Li, Zhang & Qu, 2019*).

## Correlation between the number of fixations, fixation duration, and 3-point field goal percentage

The ability to visually search and process information during the execution of a 3-point shot significantly influences the 3-point field goal percentage. Prior to initiating action responses, players must effectively select and analyze relevant visual cues (*Xi, Wang & Yan, 2004*; *Zhao, Li & Zhao, 2023*; *Jin, Ge & Fan, 2023*). In order to enhance and sustain a high 3-point field goal percentage, it is imperative to maintain a heightened level of attention and a stable fixation position. Psychologically, the key lies in directing focus towards the appropriate range during shooting (*Kevin & Dale, 2005*; *Tan et al., 2020*). There was no statistically significant difference observed in the mean number of fixations and 3-point field goal percentage participants within each area of interest (AOI) across varying levels of exercise intensity. The average front fixation duration exhibited a significant positive correlation with the 3-point field goal percentage across various exercise intensities. This suggests that in order to maintain or improve the 3-point field goal percentage, refinement of front area information processing is necessary. The present study aligns with the findings of prior research, indicating that individuals exhibiting superior shooting accuracy tend to exhibit longer fixation durations (*Walsh, 2014*; *Harle & Vickers, 2011*; *Vine, Moore & Wilson, 2014*). We believe that the fixation characteristics exhibited by basketball players during three-point shooting are intricately linked to years of rigorous training, culminating in the development of a steadfast fixation pattern through long-term systematic practice. The decrease in shooting performance during high-intensity can be attributed to the heightened attentional demands imposed on players, which disrupts their habitual fixation mode and subsequently impairs shooting accuracy. Additionally, intensified exercise leads to fatigue accumulation, thereby compromising the efficiency of automated calculation processes and further impacting shooting accuracy.

In general, moderate intensity exhibited a statistically significant positive impact on the 3-point field goal percentage, whereas both low and high intensities were associated with a decrease in the 3-point field goal percentage. This phenomenon may be attributed to the correlation between exercise intensity and cognitive arousal levels, as moderate exercise intensity optimizes arousal levels and enhances exercise performance, while low exercise intensity fails to induce optimal arousal levels (*Li, Zhang & Qu, 2019*). Conversely, high exercise intensity (heart rate > 138 BPM) hampers arousal levels and consequently impairs exercise performance (*Hu, 2014*; *Yang, Zhang & Ran, 2011*; *Tomporowski & Ellis, 1986*). Furthermore, empirical research has demonstrated that varying exercise intensities can induce alterations in the body's hormonal composition, thereby influencing cognitive task performance (*Chang & Eenter, 2009*; *Panchuk & Vickers, 2006*). By integrating psychological and physiological perspectives, we can elucidate the variations in 3-point field goal percentage performance across different exercise intensities: moderate exercise intensity can optimize arousal levels, as evidenced by an upward trend in plasma epinephrine and norepinephrine concentrations at this intensity. These hormone levels exhibit a positive correlation with cognitive performance (*Chang & Eenter, 2009*), leading to enhanced cognitive and motor abilities and improved accuracy in three-point field goal shooting. However, both low and high intensity exercise have been found to

impede cognitive arousal levels while simultaneously elevating blood ammonia and lactic acid concentrations, thereby impairing players' cognition and motor performance (*Yang, Zhang & Ran, 2011*).

## LIMITATIONS OF THE STUDY

Sample selection bias: Studies may only be conducted on specific cohorts, such as professional athletes or college students, potentially limiting the generalizability and applicability of the findings. Sample selection bias: In this type of study, it is imperative to concurrently consider multiple factors such as athletes' skill level, emotional state, and health status. Without meticulous control over these variables, ensuring the reliability of the results becomes arduous. Influence of experimental context: Conducting shooting experiments in controlled laboratory conditions differs from real-life gaming scenarios, and the lack of realism in the experimental environment may introduce a discrepancy between study findings and actual situations. Limitations of data analysis methods: The processing and statistical analysis of eye movement data may entail certain limitations. Different analytical approaches can yield divergent conclusions, necessitating the careful selection of appropriate methodologies. In future studies, it is advisable to employ rigorous sampling techniques, ensure an adequate sample size, and refine data collection methods. These measures will augment the study's scope, representativeness, and depth of analysis, ultimately enhancing the reliability and applicability of the findings.

## CONCLUSION

When executing the 3-point shot with moderate intensity, the athlete's visual fixation point becomes concentrated and stable, facilitating an efficient information search strategy, precise information processing, and ultimately yielding a heightened the 3-point field goal percentage. Variations in exercise intensity lead to an expansion of the fixation range, instability in fixation, alterations in information search strategy and processing degree, ultimately resulting in a decrease in the three-point field goal percentage. The fixation position at the front exhibited a positive impact on the 3-point field goal percentage.

### Funding
This work was supported by the 2022 China Social Science Fund Project (No. 22BTY055). The funders had no role in study design, data collection and analysis, decision to publish, or preparation of the manuscript.

### Grant Disclosures
The following grant information was disclosed by the authors:
2022 China Social Science Fund Project: No. 22BTY055.

### Competing Interests
The authors declare there are no competing interests.

## Author Contributions

- Xuetong Zhao conceived and designed the experiments, performed the experiments, analyzed the data, prepared figures and/or tables, authored or reviewed drafts of the article, and approved the final draft.
- Chunzhou Zhao conceived and designed the experiments, performed the experiments, analyzed the data, prepared figures and/or tables, authored or reviewed drafts of the article, and approved the final draft.
- Na Liu performed the experiments, analyzed the data, prepared figures and/or tables, and approved the final draft.
- Sunnan Li performed the experiments, analyzed the data, authored or reviewed drafts of the article, and approved the final draft.

## Human Ethics

The following information was supplied relating to ethical approvals (*i.e.*, approving body and any reference numbers):

This study adhered to the ethical standards for human subject research and received approval from the Institutional Review Board (IRB-20221126) at the School of Physical Education and Sports, Beijing Normal University.

## Data Availability

The raw data is available in the Supplemental File.

## Supplemental Information

Supplemental information for this article can be found online at http://dx.doi.org/10.7717/peerj.17634#supplemental-information.

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
