# Peer review of "Investigating the eye movement characteristics of basketball players executing 3-point shot at varied intensities and their correlation with shot accuracy"

_PeerJ, doi:10.7717/peerj.17634_

## Round 0.1 · original submission · Major Revisions

The study examining the eye movement characteristics of basketball players during 3-point shots requires significant revisions. Key issues include a more comprehensive and updated literature review, improved clarity and flow in the abstract, introduction, and methodology sections, and a more precise articulation of the study's significance in addressing knowledge gaps. The manuscript lacks consistency and accuracy in reporting data, particularly in the number of shots and players' shooting accuracy. Additionally, the discussion section needs a focused approach, beginning with a concise summary of findings and a direct comparison with existing literature. Theoretical relevance and additional references are required to support specific claims and enhance academic rigor. Finally, the figures and data presentation requires better organization and precision, with a thorough check and justification for any data exclusion in the findings.

In response to the reviewers' feedback, we request a major revision of your manuscript. Please address each comment thoroughly and clearly, ensuring your work aligns with the high standards of scholarly publication. Accurate and relevant citations should support your revisions. We appreciate your efforts in enhancing the manuscript and await its resubmission.

Reviewer 1 ·

Basic reporting

This study aims to examine the eye movement characteristics of high-level basketball players during the execution of 3-point shots at various exercise intensities and to explore the correlation between these eye movement characteristics and 3-point field goal percentage. These are intriguing research topics. However, I would like to highlight several areas that necessitate further clarification:

Abstract and Introduction:
• The abstract would benefit from the inclusion of key statistics, which would provide a quantifiable overview of the study's outcomes (L 19-25).
• The introduction, while covering essential topics, could be enhanced for better flow and clarity. The transition from discussing three-point shots to Visual Control Training seems abrupt (L 19-25). A more streamlined approach, perhaps starting with a broader context of eye movement in sports, would be beneficial. Additionally, integrating specific references, such as (doi 10.1016/j.physbeh.2016.08.011; 10.1016/j.bandc.2012.09.001; DOI: 10.1037/0033-2909.99.3.338), could strengthen the framing of the research problem and demonstrate the study's novelty.
• The authors fail to review existing literature on pertinent topics such as the impact of physical exertion on oculomotor functions during basketball shooting (e.g., doi 10.23736/S0022-4707.17.07522-3) or visual search (e.g., doi 10.1080/17461391.2018.1538391). This omission undermines the justification of their research's originality.
• The introduction section should be more consistent.

Clarity and Terminology:
Certain terms in the manuscript could be made clearer. For instance, the phrase "mainly focused on the front and the top" (L 292) might be better understood with additional context or explanation. Similarly, "based on deep special theoretical knowledge" (L 301) needs supporting citations for credibility. The term "The fixation system" (L 306) seems vague and could be more precisely defined. Furthermore, elaborating on the "aforementioned theory" (L 317) would provide clarity and relevance to the discussion.
Consistency and Accuracy:
There appears to be an inconsistency in the number of shots reported. The methodology describes each participant executing three shots at each intensity level, implying a total of 180 shots for 20 participants (L 146), but only 100 shots are mentioned in the analysis (L 159). Clarification on this discrepancy would enhance the manuscript's accuracy. Additionally, the authors do not provide data on the shooting accuracy of the players in the study.

Experimental design

Methodology:
• Was the number of participants sufficient to achieve adequate test power? Was a priori calculation conducted before the study?
• The separation of "Experimental Scenarios" and "Design of Experiment" into two sections seems redundant (L 128-136). A combined, cohesive section could present the methodology more effectively.
• The manuscript would benefit from a clearer articulation of how the study addresses a specific knowledge gap in the field, thereby emphasizing its significance and relevance.
• In line 94, the authors propose two hypotheses, yet they do not back these up with citations. It is unclear which previous research forms the basis for these hypotheses.

Statistical Methods:
Relocating lines 226-229 to the section on statistical methods would improve the organizational structure and readability.

Validity of the findings

Discussion and Interpretation:
The discussion section, while informative, could be more focused. The current first paragraph might be better suited to the introduction (L 19-25). Beginning the discussion with a concise summary of the key findings would set a clear foundation for further analysis. Simplifying the discussion by directly comparing the study's results with existing literature and underscoring the physiological and psychological mechanisms observed would enhance coherence. Highlighting the study's novelty and incorporating a limitations section, as well as practical implications for basketball training, are recommended.
Limitations and Implications:
Acknowledging the subjective nature of determining exercise intensity and its potential impact on the study's conclusions is crucial. Including this aspect in the limitations section would strengthen the manuscript's validity.
Theoretical Relevance:
The reference to the dual cognitive theory (L 352-353) seems tangential and not directly addressed in the study. Focusing on aspects more relevant to the research would be more beneficial.
Citations and References:
Additional references are needed for specific statements (L 325-326, 328-329) to support the claims and enhance the manuscript's academic rigor.

Additional comments

None

·

Basic reporting

Figures might be better organized:
1. Figure1 and Figure2 didn't show much information. Maybe the author can combine them and save the space for more important figures about the result.
2. Figure 3 and Figure 4 showed the details of the data listed in table 1&2, but didn't show any stats which can lead to the conclusion. I suggest the authors to organize the bars in a way that 3 intensities of each AOI stand together and label the significance by *, so that the figure can better serve the purpose of supporting discussion.
3. It's better to add a few scatter plot to show the significant correlations showed in table 4. Although I doubt at least one significant r is not correctly calculated. I'll talk about that in the 3rd section.
4. Some p values were not precisely presented, instead of just showing p<0.05. It's better to present the p value instead of its range, unless it's too small.
5. In Line 191 and 203, p=0.000 was not true. It's better to present as p<0.001.
6. Some correlation were reported wrongly. e.g. in Line 263, the authors wrote r=-0.009 but a trivially positive correlation. Also Line 269.
7. Line 267-273 is a mess. Some results were presented twice with different explanations. Please double check the data and the language, and make sure you know what you are talking about.

Experimental design

No comment.

Validity of the findings

Table 4 showed a significant negative correlation at top-right undter high intensity, whose r=-0.505. However, I checked the raw data in the supplemental materials, and found that r^2=0.0661. I guess the authors excluded one subject's data whose fixation duration at this AOI was 0 to get this significant correlation. I didn't see the reason of doing this in the methods. Please check all results presented followed the methods reported in the manuscript. The authors need to give their reason if some data need to be excluded.

Additional comments

In this study, the authors investigated high level basketball players’ eye movement characteristics at different exercise intensities, and correlated that with the players’ accuracy of 3-point shot. The story is simple and clear. The methods and results are clearly presented and are discussed in depth. A few concerns or suggestions were listed above.

·

Basic reporting

.

Experimental design

.

Validity of the findings

.

Additional comments

The manuscript needs a intensive revision. The following comments must be addressed in further round:
1. The literature is very poor. The authors needs to work on lastest publications and add papers from 2020-2023 and cite them.
2. The methodology must be in a flow. Current methodology is not clear. It has many flaws. It is better to mention an algorithm or pseudo code to defend methodology.
3. The discussion must be added and the authors are requested to explain every figure and table in discussion section.

---

## Round 0.2 · Minor Revisions

The manuscript requires targeted revisions for clarity and cohesion, particularly in the introduction, which suffers from redundancy and a lack of logical flow. Specifically, discussions on eye movement definitions and technologies (lines 35-36 and 115-116) should be consolidated, and related content between lines 75-83 and 90-114 needs unification. Additionally, the narrative should map out the research landscape, knowledge gaps, and the study's unique contributions without revisiting covered topics. The reference list and in-text citations need correction for accuracy, mainly using surnames instead of first names. Obvious errors include incorrect statistical notation (p<0.000) at lines 226 and 289, misuse of "vision system" (should be "visual system") at line 324, inaccuracies in table title serial numbers, and miscellaneous language mistakes. A thorough review and correction of these issues will significantly enhance the manuscript's quality.

Reviewer 1 ·

Basic reporting

The introduction section is overly lengthy and lacks cohesion. Specifically, themes introduced in lines 115-116 are redundantly revisited from earlier discussions in lines 35-36. It is recommended that these sections, focusing on eye movement definitions and the employed technologies for their investigation, be consolidated. Additionally, the content between lines 75-83 and 90-114, which addresses related issues, should be unified. The narrative structure of the paragraphs needs a logical flow, avoiding the reiteration of previously covered topics. It should clearly delineate the existing research landscape, identify the gaps in current knowledge, and highlight the unique contributions of the authors' research, thereby underscoring its originality and relevance.
The reference list requires thorough correction, both in terms of citation in the text and the accuracy of the citations themselves: in many cases, the authors use first names instead of surnames in references. This also requires correction in the reference list: Here are some examples:

Line 36 is (Kathy, Sanchez & Carrol, 2020) but should be (Conklin, Pellicer-Sánchez, Carrol, 2020).
Line 82 is Teresa et al. 2018, but should be Zwierko et al. 2018. See Zwierko, T., Popowczak, M., Woźniak, J., & Rokita, A. (2018). Visual control in basketball shooting under exertion conditions. The Journal of Sports Medicine and Physical Fitness, 58(10), 1544–1553. https://doi.org/10.23736/S0022-4707.17.07522-3
Line 56 is Frank J H L, Diane L G, Lee Y C, Chiu Y H, Sean L, Liu H Y. 2020 but should be Lu, Gill, Lee, etc.
Line 90 is missing a reference year.
Line 108 is Terry 2016, but should be McMorris 2016. See: McMorris T. (2016). Developing the catecholamines hypothesis for the acute exercise-cognition interaction in humans: Lessons from animal studies. Physiology & behavior, 165, 291–299. https://doi.org/10.1016/j.physbeh.2016.08.011
In the reference list is “Teresa Z, Wojciech J, Beata F, Miłosz S, Rafał B, Dorota K N. 2018. Oculomotor dynamics in skilled soccer players: The effects of sport expertise and strenuous physical effort. Eur J Sport Sci19 (5): 612-620. DOI: 10.1080/17461391.2018.1538391” It should be: Zwierko, T., Jedziniak, W., Florkiewicz, B., Stępiński, M., Buryta, R., Kostrzewa-Nowak, D., Nowak, R., Popowczak, M., & Woźniak, J. (2019). Oculomotor dynamics in skilled soccer players: The effects of sport expertise and strenuous physical effort. European journal of sport science, 19(5), 612–620. https://doi.org/10.1080/17461391.2018.1538391 - Additionally, this item is missing in the text and should be included in the paragraph from lines 75-83 and/or in discussion section.
Line 129 – still lacks source references for the adopted hypothesis "Based on previous research…" specify which previous studies.
Lines 141-143 – do not mention the statistical test on which the a priori calculation was based, this should be supplemented.
In Table 1 and Table 4, instead of 0.000 use < 0.000.

Experimental design

no comment

Validity of the findings

no comment

Additional comments

no comment

·

Basic reporting

The author corrected most of the mistakes I pointed out last time, except the following:
L226 (in the .pdf manuscript): p<0.000
L289: p<0.000
L324: vision system should be visual system
The serial numbers in the titles of the tables were all wrong.
There are also a few language mistakes. Please check the whole manuscript.

And I'm just curious, is there any difference between address 1, 2 and 4? Seems to be the same college.

Experimental design

No comment.

Validity of the findings

Well, as impact is not my concern as a reviewer, no comments.

Additional comments

No comment.

---

## Round 0.3 · Minor Revisions

Thank you for your diligent efforts in revising the manuscript. The changes made thus far have significantly contributed to the advancement of your paper. However, upon further review, there are some remaining comments from R1 which still need to be addressed to ensure the manuscript meets our publication standards. Please see the detailed comments from the reviewer and revise your manuscript accordingly.

We look forward to receiving your revised submission.

Reviewer 1 ·

Basic reporting

The authors have revised the manuscript according to most of the reviewer's comments. Nevertheless, significant inaccuracies persist in the document, necessitating further corrections.

- Line 81 incorrectly cites Zwierko et al. as (2019) when it should be (Zwierko et al., 2018).
- Lines 117-118 need to be updated with the necessary information: G*Power 3.1.9.7 (Heinrich Heine Universität Düsseldorf, Düsseldorf, Germany) (Faul et al., 2007). Please specify for which method the effect size and the level of significance at 0.05 were determined.
- Line 131 lacks source data for Tobii Glasses 3.
- The manuscript is missing a section on the study's limitations, which should be included.
- Line 528 The reference to Zwierko et al., 2019 is not included in the text of the manuscript. This should be addressed.

Experimental design

no comment

Validity of the findings

The manuscript is missing a section on the study's limitations, which should be included.

Additional comments

no comment

·

Basic reporting

No more comment.

Experimental design

No comment

Validity of the findings

No comment

Additional comments

No comment

·

Basic reporting

The manuscript is accepted for publication.

Experimental design

The manuscript is accepted for publication.

Validity of the findings

The manuscript is accepted for publication.

---

## Round 0.4 · Minor Revisions

Thanks to the authors for the revision. However, several comments for further revision are recommended:

1. The revision process is cumbersome due to the lack of highlighted texts in the amendments. Please ensure changes are clearly marked to facilitate a smoother review process.

2. In the abstract, remove unnecessary details such as the Tobii version and statistical values. Aim for a concise and informative abstract.

3. Line 115 does not fully address Reviewer 1's comment regarding the citation of G*Power 3.1.9.7 (Germany). Please revisit the source page, cite appropriately, and give proper credit to the G*Power authors. [G*Power source](https://www.psychologie.hhu.de/arbeitsgruppen/allgemeine-psychologie-und-arbeitspsychologie/gpower)

4. Line 131 lacks sufficient sourcing for the Tobii Glasses 3 data. Please specify the source and detail the data processing steps. [Refer to this article for guidance](https://link.springer.com/article/10.3758/s13428-019-01314-1)

5. Ensure unity between in-text citations and the reference list. As noted by the reviewer, some references cited in the text are missing from the list. Please review the manuscript thoroughly. It is strongly recommended to follow the PeerJ guidelines for publication and style. [PeerJ Style Guidelines](https://peerj.com/about/author-instructions/#style-considerations)

6. The newly added section on limitations currently lacks depth and fails to substantively address the study's constraints compared to existing literature. It is essential to not only mention apparent limitations but also to explore methodological challenges such as sampling errors and insufficient sample sizes that can lead to selection bias and compromise the validity of statistical conclusions. The scarcity of prior research on the topic underscores the need to thoroughly evaluate literature gaps, which could guide the development of innovative research frameworks. Additionally, a critical assessment of data collection methods should be undertaken to pinpoint any potential limitations in analysis. For future studies, it is recommended to employ rigorous sampling techniques, ensure adequate sample sizes, and refine data collection methods. These measures will enhance the research's scope, representation, and analytical depth, ultimately strengthening the reliability and applicability of the study's findings. This more comprehensive approach to discussing limitations should be positioned thoughtfully within the manuscript, ideally after the discussion but before the conclusion, to maintain logical coherence and flow.

---

## Round 0.5 · accepted · Accept

I was asked to step in, as the previous Editor is out of contact.

I have read all revised manuscript and compared to original manuscript I am happy with the revisions and happy to Accept this manuscript